# Allogenic Adipose-Derived Stem Cells in Diabetic Foot Ulcer Treatment: Clinical Effectiveness, Safety, Survival in the Wound Site, and Proteomic Impact

**DOI:** 10.3390/ijms24021472

**Published:** 2023-01-12

**Authors:** Beata Mrozikiewicz-Rakowska, Ilona Szabłowska-Gadomska, Dominik Cysewski, Stefan Rudziński, Rafał Płoski, Piotr Gasperowicz, Magdalena Konarzewska, Jakub Zieliński, Mateusz Mieczkowski, Damian Sieńko, Tomasz Grzela, Maria Noszczyk, Barbara Paleska, Leszek Czupryniak, Malgorzata Lewandowska-Szumiel

**Affiliations:** 1Department of Endocrinology, Centre of Postgraduate Medical Education, Bielanski Hospital, 01-809 Warsaw, Poland; 2Laboratory for Cell Research and Application, Medical University of Warsaw, 02-097 Warsaw, Poland; 3Clinical Research Centre, Medical University of Bialystok, 15-276 Bialystok, Poland; 4Institute of Biochemistry and Biophysics, Polish Academy of Sciences, 02-106 Warsaw, Poland; 5Department of Medical Genetics, Warsaw Medical University, 02-106 Warsaw, Poland; 6Department of Forensic Medicine, Warsaw Medical University, 02-007 Warsaw, Poland; 7Interdisciplinary Centre for Mathematical and Computational Modelling, University of Warsaw, 02-106 Warsaw, Poland; 8Department of Diabetology and Internal Diseases, Medical University of Warsaw, 02-097 Warsaw, Poland; 9Doctoral School, Medical University of Warsaw, 02-091 Warsaw, Poland; 10Department of Histology and Embryology, Medical University of Warsaw, 02-004 Warsaw, Poland; 11Melitus Aesthetic Medicine and Anti-Aging Clinic, 01-627 Warsaw, Poland

**Keywords:** cell therapy, diabetic foot ulcer, adipose-derived stem cells, ADSC, ATMP, proteomics, clinical trial, allogenic cell therapy

## Abstract

Although encouraging results of adipose-derived stem cell (ADSC) use in wound healing are available, the mechanism of action has been studied mainly in vitro and in animals. This work aimed to examine the safety and efficacy of allogenic ADSCs in human diabetic foot ulcer treatment, in combination with the analyses of the wound. Equal groups of 23 participants each received fibrin gel with ADSCs or fibrin gel alone. The clinical effects were assessed at four time points: days 7, 14, 21 and 49. Material collected during debridement from a subset of each group was analyzed for the presence of ADSC donor DNA and proteomic changes. The reduction in wound size was greater at all subsequent visits, significantly on day 21 and 49, and the time to 50% reduction in the wound size was significantly shorter in patients who received ADSCs. Complete healing was achieved at the end of the study in seven patients treated with ADSCs vs. one treated without ADSCs. One week after ADSC application, 34 proteins significantly differentiated the material from both groups, seven of which, i.e., GAPDH, CAT, ACTN1, KRT1, KRT9, SCL4A1, and TPI, positively correlated with the healing rate. We detected ADSC donor DNA up to 21 days after administration. We confirmed ADSC-related improvement in wound healing that correlated with the molecular background, which provides insights into the role of ADSCs in wound healing—a step toward the development of cell-based therapies.

## 1. Introduction

Diabetic foot ulcer (DFU) is one of the most common complications of poorly controlled diabetes mellitus [1]. Despite many efforts, the search for an effective therapy in the healing of DFU remains a very topical clinical issue and one of the major problems in the field of chronic wounds [2,3]. Cell-based therapies using mesenchymal stem cells (MSCs) are of high interest due to their pro-regenerative and immunomodulatory properties [4]. MSCs were discovered in the bone marrow [5,6]; currently, they are obtained from various tissues, e.g., adipose tissue, umbilical cord, and even menstrual blood [7,8]. Adipose-derived stem cells (ADSCs) are one of the most common materials used for MSC therapy, as they are abundant, easy to obtain and have high immunomodulatory potency [4,9,10,11,12]. More than 300 clinical trials using ADSCs can be found at clinicaltrials.gov, while there is only one ADSC-based medicinal product registered in the EU as of now. There is some evidence that ADSCs can support wound healing through anti-inflammatory and proangiogenic effects and can enhance granulation tissue formation [9,13].

In 2019, Moon reported the successful treatment of DFU using allogenic ADSCs in a clinical trial. ADSCs were administered in the form of a hydrogel patch as a primary dressing, while in the control arm, only a secondary dressing was applied [14].

Encouraged by these results, we took the next step toward confirming and understanding the role of ADSCs as the active substance in the treatment of DFU. We also used allogenic ADSCs, though in our study design, the addition of these cells was the only variable examined. Furthermore, we tested for the presence of donor cells in the wound at subsequent time points and monitored molecular markers of ADSC-modulated wound healing by means of proteomic analysis of the recipient wound site. Currently, our knowledge about supportive MSC healing properties originates mainly from in vitro observations and animal studies. Therefore, we expect our approach to bring us closer to elucidating the mode of action of ADSCs in DFU treatment.

## 2. Results

### 2.1. Clinical Outcomes

Among 58 screened patients with neuropathic DFU (grades IA and IIA, according to the University of Texas classification), 47 individuals fulfilled the inclusion/exclusion criteria and were allocated to a study group (24 vs. 23 patients). In one patient initially allocated to the ADSC group, a small wound appeared on the contralateral leg 5 days after the application of ADSCs. This adverse reaction was assessed and determined as ADSC-independent. Following the exclusion criteria, this patient was excluded from the study group and was not included in the group statistics. Hence, both groups finally contained 46 patients, 23 each, who completed all planned medical procedures, including follow-up assessments. Notably, despite nonrandomized patient allocation, the study groups did not differ significantly in demographic or clinical baseline parameters (see Table 1). Moreover, they did not differ in wound area on the day of administration of fibrin gel with or without ADSCs (day 0) (average wound size 2.68 cm^2^ vs. 2.72 cm^2^, *p* = 0.42).

The time to 50% reduction in the size of the wound in the group receiving fibrin gel was 25.5 ± 4.2 days and in the ADSC-group it was 17.6 ± 1.5 days (*p* = 0.029). Figure 1a compares the time of wound area reduction based on the changes in wound size measured on a weekly basis. Additionally, photographic documentation was taken to illustrate the changes in patients wounds. Figure 1b shows exemplary photographs of wounds in the studied groups (for photographic documentation from the complete set of patients see Appendix A). The wound size quotient at subsequent visits was smaller in the ADSC group than in the gel group. The differences were not significant on day 7 and 14 (*p* = 0.109 and 0.055, respectively), while they were significant (*p* = 0.019 and *p* = 0.023) on day 21 and 49. The results are presented in Figure 1a. At the end of the study the wound completely healed in seven patients from the ADSC group and in only one patient from the gel group. The evaluation of pain sensation did not differ significantly between the groups (Appendix A). The lack of differences in pain between the study groups indicates that patients were appropriately selected for the study groups due to the inclusion criteria—all were characterized by an established form of neuropathy. The occurrence of pain in such a patient could be indicative of a significantly increased inflammatory process, which was an exclusion criterion for the study. As for the possible effect of ADSCs on nerve regeneration, at this stage of neuropathy, the reversal of damage to the vasa nervorum, and therefore nerve fiber reinnervation, has not yet been described in humans, and further research is needed. For example, analysis of the SDF-1/CXCR4, a pathway important for the process of neovascularisation, which is of critical importance for neuroregeneration in DFU, may be taken into account, as well as its changes under the influence of stem cells and factors secreted by them [15,16].

The groups had similar rates of adverse events (Appendix A). None of them were related to the medical procedures that the patients underwent. No serious adverse events were observed.

In general, the ADSC group benefited from the use of stem cells in terms of accelerated wound healing and a had a greater likelihood of complete wound healing.

### 2.2. Proteomic Analysis

We identified and quantified over 300 proteins in wound scrapings from each of the three LC–MS/MS groups. The samples were characterized by high heterogeneity and a wide range of protein concentrations. Sets 2 and 3 were merged for analysis. In the ADSC group, 34 proteins statistically significantly differentiated the day 0 and day 7 time points (*p* < 0.05, Benjamin-Hochberg correction) (Figure 2a–c). The identified enriched proteins are functionally connected (Appendix A). The Reactome platform identified several processes, such as antimicrobial peptide (AMP) synthesis (Appendix A), integrin signaling, and amyloid formation (Appendix A) [17]. We further compared the upregulated proteins with the course of the treatment in the ADSC group patients and found that seven of the identified proteins—GAPDH, TPI1, CAT, SCL4A1, ACTN1, KRT9 and KRT1 positively correlated with the wound closure rate (Appendix A).

Summing up, we not only discovered a panel of proteins which differentiated wound scrapings from the ADSC group and fibrin gel group, but we also found out that seven of them were positively correlated with the progress of healing.

### 2.3. Donor DNA in Wound Samples

Wound scraping samples for donor DNA admixture assessment were collected from 21 patients from the ADSC group at three time points—day 7, day 14 and day 21. Of all 63 recipient wound scraping samples (three time points per patient) from the ADSC group, donor DNA was detected in eight and nine samples using STR and ADS, respectively. Both methods detected the admixture in the same 8 scraping samples, while one more sample was positive by ADS. Donor DNA was mainly found in samples from day 7 (in 7 out of 21 patients), but single samples from day 14 and day 21 time points were also positive. The amount of donor DNA ranged from 1.4 to 4.7% of the total DNA. In the two scraping samples, where donor DNA was detected on day 14 (patient 1ABC01019) or on day 21 (patient 1ABC01027), the amount present was similar to (for 1ABC01019) or higher than (for 1ABC01027) that in the same patient on day 7 (Appendix A). However, no donor DNA was detected in samples from patient 1ABC01027 collected on day 14. An analysis of the amount of donor DNA detected through STR loci with relatively long and short amplicon sizes did not show consistent differences (Appendix A). Only in one sample (sample 1ABC01019, day 14) did we find a significant difference (*p* = 0.0041), indicating a higher result with the short loci (4.6%) than the long loci (1.9%). In other samples the trends were mixed. In four samples there was a trend for higher values at the short loci; in two samples the opposite was true; and in one sample the values were virtually identical. When samples were pooled, there was no significant difference between the mean values for the short and long loci (0.042 vs. 0.038, respectively, *p* = 0.28, for raw data see Appendix A).

Taken together, by using two independent methods, we were able to confirm the presence of donor DNA in the samples of the treated wounds. Therefore, we have shown that ADSCs are capable of migrating from the fibrin gel into the wound bed. Moreover, the STR analysis results are compatible (although they clearly do not constitute proof) with the presence of at least some living donor cells at the time of sampling.

## 3. Discussion

DFU poses a serious clinical threat, so there is a need for new therapeutic approaches, such as advanced wound healing technologies that harness the pro-regenerative and immunomodulatory potential of ADSCs. We were inspired by Moon et al., who provided evidence of allogenic ADSCs’ safe and effective use in the treatment of DFU when applied to the wound in a hydrogel patch as a primary dressing [14]. In our study, fibrin gel was administered with or without ADSCs in suspension, allowing us to confirm Moon’s findings under an experimental setup in which the presence of cells, as an active substance, was the only variable. This design was necessary to distinguish their role in wound healing. Our observations also differ from Moon’s study in that Moon et al. used cells frozen in a gel dressing, whereas in our work, cells were frozen, thawed and cultured before application in the fibrin gel. Therefore, we were able to examine cells’ viability before application, thus providing better control over the medicinal product’s quality as well as the similarity of subsequent product batches. Moreover, Moon et al.’s control group was treated solely with a polyurethane foam dressing, whereas in our work the same fibrin gel was applied in both groups. Taking into account the composition of the hydrogel used by Moon et al. was not provided, the influence of the hydrogel itself on the healing process cannot be excluded. Thus, our observation is a step forward. It supports Moon’s observation of the safety of using allogenic ADSCs in DFU—no adverse effects were connected with the procedure. We also demonstrated ADSC efficacy in terms of the time needed to reduce the wound area by 50% from the initial size, the wound size decrease at each visit, and the number of patients who achieved fully closed wounds within the time frame of the study. These findings were obtained from a nonrandomized and nonblinded study, which is a limitation of our work, even though the groups being compared did not differ significantly in demographic or clinical parameters at baseline.

On the other hand, while Moon focused on the clinical effect, we also tried to get closer to assessing the impact of the cells themselves at the molecular level. Our donor ADSC detection in the wound site as well as our proteomic analysis of wound scraping samples are novelties of this work. The presence of the ADSC donor’s genetic traces was assessed by two independent methods: STR and ES/ADS. Both methods showed that the DNA traces of donor cells were found even 21 days after administration of the medicinal product—in this case, in one sample, patient 1ABC01027 (see Appendix A). To date, cell viability after ADSC administration has been observed only in animal models [18,19], but our results show that ADSCs migrated from the fibrin gel into the wound site. Moreover, the low degradation of STR markers suggests that the DNA traces found in the wound scraping samples up to 21 days after application could come from viable ADSCs. Our observations confirm the worth of ADSC therapy and suggest that it may exert a direct impact on the course of wound healing in DFU even 21 days after application. Although we found ADSC donor genetic traces in only about 30% of patients at different time points after application, mainly on day 7, one should keep in mind that they were observed in wound scraping samples obtained during routine wound cleanup. While these procedures are meant to only remove fragments of unhealed tissue, we assume that ADSCs could have accumulated in healthy tissue that had already undergone epithelialization.

In fact, if we didn’t find traces of the donor material in the scrapings at all, it still would not rule out the presence of donor material in the healed part of the wound. Interestingly, in patient 1ABC01027, where donor DNA was detected on day 21, there were no donor DNA traces found in the sample from day 14. It seems possible that ADSCs were present in the healing part of the wound on day 14 and then migrated from there to become detectable in scrapings on day 21. Taking into account the low degradation of STR markers, we may assume that the detected cells were alive and, as such, capable of proliferation and migration. This would explain the similar or even higher amounts of DNA detected in some cases at later time points compared to earlier ones (patients 1ABC01019 or 1ABC01027). Nevertheless, such interpretations are only speculative. What we really proved was the presence of ADSCs donor DNA, at various time points from application, in human clinical observation, which is an important contribution to our knowledge of cell-based therapies.

We identified 34 proteins that were highly overrepresented or identified exclusively in samples from day 7 after ADSC application, compared to day 0 (before application). We observed elevated levels of APOA1 and PFN1 proteins in the ADSC group, proteins reported to play a role in proangiogenic processes, either by facilitating vascular endothelial cell migration (PFN1) [20] or the promotion of proliferation and differentiation of human endothelial progenitor cells (APOA1) [21]. Interestingly, increases in the levels of GAPDH, CAT, ACTN1, KRT1, KRT9, SCL4A1, and TPI correlated with the rate of wound healing by the individual patient (Figure 2d and Appendix A). The immunomodulatory characteristics of GAPDH [22] and CAT proteins have been shown in humans and in animal models, respectively [23]. These proteins are involved in macrophage polarization from the M1 to the M2 phenotype and in the inflammation-to-proliferation phase shift in wound healing, a process crucial to wound healing progress.

The overrepresentation of KRT1 and KRT9, which are specific to keratinocytes, indicates ongoing regenerative processes, as keratinocytes are not present in the native chronic wound bed [24]. ACTN1 regulates the directional migration of keratinocytes [25]. The increase in these molecules might be considered a molecular marker of re-epithelialization. It is noteworthy that the increase in the levels of these three proteins correlated positively with the wound closure rate in the ADSC group. This finding might suggest that KRT1, KRT9, and ACTN1 could become prognostic markers for ADSC-based therapies in DFU treatment. This possibility should be tested in future validation studies.

To better understand the ADSCs’ influence on wound healing, we performed a pathway analysis with the Reactome platform. We focused on the highest-probability identified pathways (*p* < 0.0001, FDR 0.01%). A number of processes might be connected to the healing process (e.g., amyloid fiber formation, plasma lipoprotein remodeling, integrin signaling). Others, such as antimicrobial peptides, play a supportive role in the process, e.g., modulating chronic inflammation, mainly by wound microbiome control [26]. Sequential activation of the processes that must occur in the wound healing process is, in our opinion, additional proof that the obtained data are not random findings. Interestingly, 40% of proteins observed in increased levels have or are predicted to have AMP properties. Five proteins were already annotated in databases as AMPs (Figure 2d), and nine others were highly scored by AmpGram to have AMP probability [27] (Appendix A). Keeping potentially harmful microbiota at bay may be one of the factors that enhance wound healing [28,29]. These findings show the variety of wound healing-accelerating effects generated by ADSCs.

## 4. Materials and Methods

### 4.1. Clinical Observation Design

The study was performed under hospital exemption in patients with DFU of neuropathic origin for whom standard wound healing treatment had failed. It was a prospective parallel-group trial without randomization and without blinding. Patient preselection was based on the inclusion/exclusion criteria described below. This clinical observation was performed with a strict adherence to “STROBE” (Strengthening The Reporting and OBservational studies in Epidemiology) guidelines.

The patient inclusion criteria were as follows: age >18; chronic wound area of 1–25 cm^2^; glycated hemoglobin (HbA1c) <11%; transcutaneous oxygen tension (tPO2) ≥ 30 mmHg in the wound area or systolic pressure on the distal tibial arteries ≥50 mmHg; and general condition of the patient that in the opinion of the investigator allowed participation in all study procedures.

The patient exclusion criteria were ulcer etiology other than DFU; wound area >25 cm^2^, HbA1c ≥11%; clinically significant limb ischemia as shown by tPO2 <30 mmHg or arterial pressure on the distal tibial arteries <50 mmHg; active wound infection; allergy to thrombin; active venous thrombosis; presence of the wound on the contralateral foot; any systemic disease in the stage of exacerbation (acute or uncontrolled); and any oncological treatment within the last 5 years.

The early study termination conditions were as follows: the withdrawal of the patient’s consent to participate in the study, significant violations of the study protocol by the patient or investigator, and any condition where (upon individual decision of the investigator) further participation in the study could pose any risk to the patient.

Patients who fulfilled all inclusion criteria and did not meet any exclusion criteria were allocated to one of two study groups as follows. The first, called the fibrin gel group, was treated with fibrin gel covered with a secondary dressing, and the second, the ADSC group, was treated with ADSCs suspended in fibrin gel and secured with a secondary dressing. Although the study was not randomized, we confirmed that the two groups were similar at baseline. After the study, demographic and clinical parameters were compared to detect any differences that might have arisen between them.

The volunteers gave their informed consent to participate in the trial. The Bioethics Committee at the Medical University of Warsaw approved the trial (approval no. KB/128/2019 and KB/3/A2021), and the trial was conducted according to the ethical guidelines of the World Medical Association Declaration of Helsinki. The clinical part of the study was registered at ClinicalTrials.gov (identifier: NCT03865394) and ran from October 2019 to July 2021 in the Department of Diabetology and Internal Diseases of the Medical University of Warsaw.

The clinical outcomes were the wound healing rate and the safety of the applied therapy. The former was assessed as the time needed to reduce the wound area by 50% from the initial size, determined by calculating the relative size of the wound at each visit to the size on the day of application of fibrin gel with or without ADSCs, and the number of patients who achieved a fully closed wound within the time frame of the study. Molecular assessment of the therapeutic efficiency was accomplished by analyzing the expression of selected proangiogenic and immunomodulatory factors and detecting donor DNA traces in the wounds of both study groups.

### 4.2. ADSC Collection and Preparation

ADSCs from one healthy donor were used in order to avoid any donor-related variability in results. The informed consent of the donor, approved by the Bioethics Committee (opinion nos. KB/128/2019 and KB/3/A2021), was obtained. Medical qualification of the donor was based on the rules determined by a transplantation law. ADSCs for clinical application were produced in the Laboratory for Cell Research and Application, Medical University of Warsaw, in compliance with the GMP standard laid down for Advance Therapy Medicinal Products (ATMP) with the consent of the Chief Pharmaceutical Inspectorate (consent no. 01/0556/2019). Lipoaspirate obtained from the abdominal adipose tissue at the Mellitus Clinic was used as a starting material. After isolation, cells were cultured to obtain MSCs and stored frozen in liquid nitrogen. Prior to application, the cells were thawed, briefly cultured and then used up to the third passage.

For the ADSCs isolation, collagenase NB 6 GMP Grade (Nordmark Pharma GmbH, Uetersen, Germany) solution was used in the proportion of adipose tissue to collagenase equal to 1:1 (*v/v*). The collagenase treatment was carried out for 1 h at 37 °C under constant shaking. Prior to digestion, the adipose tissue was washed extensively with 1% antibiotic-antimycotic solution (Corning Inc., Corning, NY, USA) in phosphate-buffered saline (PBS) (ThermoFisher Scientific, Waltham, MA, USA). To get rid of the blood cells, ZAPR™ (Incell, Frisco, TX, USA) a buffer for lysis of erythrocytes, was used according to the manufacturer’s recommendations. The cells were then separated by centrifugation (at 300× *g* for 5 min at RT), rinsed with fresh NutriStem^®^ XF Basal Medium, and filtered through a cell strainer. After isolation cells were counted with an ADAM-MC automated cell counter (NanoEnTek Inc., Seoul, Republic of Korea), plated in T75 culture flasks and cultured at 37 °C and 5% CO_2_ in a humidified atmosphere in the complete culture medium MSC NutriStem^®^ XF Basal Medium with Supplement Mix (Biological Industries, Beit-Haemek, Israel) and 0.1% antibiotic-antimycotic solution (Corning Inc., Corning, NY, USA). MSC NutriStem^®^ is a xeno-free and serum-free medium devoted to translational use for cell cultures carried out under the Good Manufacturing Practice (GMP) regime. Isolated cells were expanded in culture using the same conditions. Cells from the first and the second passage were stored frozen in liquid nitrogen. 

In order to confirm the desired immunophenotype of the cells, a common set of positive and negative cell surface markers were determined by means of cytometric analysis [30]. Conjugated mouse monoclonal antibodies directed against CD29, CD31, CD34, CD45, CD73, CD105 and CD146 (ThermoFisher Scientific, Waltham, MA, USA) were used, and the analysis was performed by using a CytoFlex Beckman Coulter flow cytometer. More than 98% of the population was CD105+, CD73+ and CD29+, CD31−, CD45−, and CD146−, and more than 90% of the cells were CD34-. Overall, FACS analysis revealed that 98.66% of the total population of analyzed cells fulfilled the minimal criteria for ADSCs.

### 4.3. Manufacturing of the Medicinal Product

Prior to application, the cells were thawed, cultured in complete MSC NutriStem^®^ XF Basal Medium with Supplement Mix (Biological Industries, Beit-Haemek, Israel) and 0.1% antibiotic—antimycotic solution (Corning Inc., Corning, NY, USA) until appropriate confluence (2–3 days cultured) and passaged—only cells from the second or third passage were used for the final product. The cells were then detached using StemPro™ Accutase™ Cell Dissociation Reagent (ThermoFisher Scientific, Waltham, MA, USA). The cells were then rinsed twice with sodium chlorate (Fresenius Kabi, Warsaw, Poland), and they were centrifuged at 350× *g* for 5 min at 22 °C after each rinse. Cell number and viability were then assessed (automated cell counter ADAM-MC, NanoEnTek Inc., Seoul, Republic of Korea). Finally, each individual batch released to the clinic consisted of a 1 ml suspension of 2.5 × 10^6^ ADSCs in sodium chloride with a cell viability greater than 70%.

Additionally, to assure patient safety, samples were taken several times during the manufacturing process for analysis to confirm sterility and to test the level of endotoxins according to the GMP rules specific to ATMPs.

### 4.4. Clinical Procedures

At the screening visit (day-7), all eligible patients were subjected to verification of the inclusion/exclusion criteria. The main criteria for participation in the study included wound size, assessed using a Silhouette^®^ laser scanner; tPO2, measured by a TCM400 device; systolic pressure in the distal limb arteries; and other clinical features described in the study protocol. Furthermore, blood samples were taken for basic laboratory and viral tests. The same procedures were repeated during a visit on day 0 before the application of the tested formulation to the wound.

In the ADSC group, the cell suspension in fibrin gel was applied directly onto the wound using a special applicator. In order to ensure the required quality of the fibrin gel, the commercially available medicinal product TISSEEL Lyo (Baxter, Deerfield, IL, USA) was used. According to the manufacturer’s instructions, the two components of the product, i.e., the Sealer Protein (Human Fibrinogen and Aprotinin) Solution and Thrombin Solution can be applied using a Duploject system, which allows for the combination of the two components during application. For the ADSC group, cell suspension in NaCl 0.9%, as described above, was added to each component of the system in equal proportions, mixed, and applied to the wound using a Duploject system. For the patients who received fibrin gel only (Fibrin gel group), the same procedure was used with the addition of NaCl 0.9%, without cells as the only difference. For wounds up to 12 cm^2^, 2.5 *×* 10^6^ cells in 1 ml of fibrin gel were used, whereas for wounds above 12 cm^2^, the volume of cell suspension was increased proportionally. Patients were administered this formulation only once, during a visit on day 0. The wound was then covered with a secondary dressing (UrgoTul, Laboratoires Urgo, Chenôve, France). That dressing was changed every 3 ± 1 days, and control hospital visits were every 7 ± 3 days. The treatment efficacy was assessed at weekly intervals (days 7, 14, 21) and on day 49. The same procedures, except for the ADSC admixture to the fibrin gel suspension, were applied in the gel group (Figure 3).

Patient participation in the study was terminated when the wound was healed or 49 days after the application, whichever came first. If the wound was not completely healed 49 days after application, the patient was further treated in compliance with the general principles of chronic wound care.

### 4.5. Evaluation of the Clinical Outcomes

The clinical effectiveness and safety of treatment were assessed at each visit through the following:measurement of the wound surface;local and systemic tolerance of therapy, including the episodes of wound infection or any other adverse events;number of patients in each group who achieved complete wound closure;evaluation of the pain scale: evaluation of the changes in pain experienced by the patient due to the wound treated with the tested method, in relation to the wound treated in a standard way through the use of the visual analog scale (VAS).

### 4.6. Wound Debridement

According to the TIME recommendations, to remove any unvital or contaminated material, the wound was cleaned up using a Volkmann spoon curette during each visit until the wound was completely healed. Part of this waste material (hereafter, scrapings), particularly those removed from the bottom of the ulcer, was deep-frozen and stored at −80 °C until further molecular assessments were undertaken. The above debridement was not performed after the wound had completely healed.

### 4.7. Proteomic Analysis

Each wound scraping sample was dissolved in a lysis buffer, denatured and sonicated. Debris was removed by centrifugation. Proteins were then precipitated with a modified methanol/chloroform protocol [31].

The pellet of extracted proteins was digested with LysC/trypsin as described [32] and labeled with iTRAQ8-plex. Samples were organized into three sets for LC–MS/MS analysis: set 1: scrapings samples of three patients from the fibrin gel group at two time points (day 0, day 7); sets 2 and 3: scrapings samples of six patients from the ADSC group at two time points (day 0, day 7) analyzed as two separate groups of three. Combined samples were prefractionated and analyzed by LC–MS/MS online UPLC and a QExative Orbitrap mass spectrometer at the Institute of Biochemistry and Biophysics PAS. Data were searched using the MaxQuant platform against UniProt reference proteome database taxonomy for statistical analysis with Scaffold Q+S. For the detailed protocol, see Description S1. Data were deposited in the PRIDE repository under access number PXD032099.

### 4.8. Search for Donor DNA in Wound Scrapings

Seventy wound scraping samples were analyzed (63 samples collected from 21 recipients in the ADSC group at three time points (day 7, day 14, day 21)), six negative controls collected from two recipients in the gel group and one positive control (donor). We searched for donor DNA in wound samples by two methods: (i) short tandem repeat (STR) analysis; and (ii) amplicon deep sequencing (ADS) designed to contain a rare variant (A/G at rs35874463) selected after exome sequencing (ES) of the donor. The size of donor DNA admixture was estimated as a percentage of total DNA in the scraping samples by measuring the areas under the peak (STR analysis) or as the proportion of variant reads (amplicon sequencing). Donor DNA degradation was assessed by analyzing whether the amount of donor DNA detected was higher when shorter vs. longer STR alleles were considered. Detailed methods are described in Description S2.

### 4.9. Statistical Analyses

To compare the ADSC and fibrin gel groups before treatment, the Wilcoxon test (for quantitative variables) and Fisher’s exact test (for categorical variables) were used. The group comparison results are summarized in Table 1.

Significantly changed proteins in the proteomics experiment were those with *p* < 0.05 in the Mann–Whitney test after Benjamini–Hochberg correction.

To minimize the effect of the initial wound size variance, for each patient the wound size measured at subsequent visits was divided by the wound size during the application of treatment, yielding a relative wound size (wound size quotient). The quotients on day 7, day 14, day 21, and day 49 are named Q1, Q2, Q3 and Q4, respectively. These new variables were used in several statistical analyses:To compare the healing rate in both groups, the Wilcoxon-Mann–Whitney test was used;Exponential regression was used to determine the half-time of wound surface decline;A Spearman correlation was used to evaluate the relationship between the rate of healing and upregulated protein expression. The correlations were calculated between the relative wound size and a measure of increase in protein expression. This measure was defined as the difference in the logarithm of the protein intensities from the mass spectrometry measurements. The measurements were performed in weeks 0 and 1.

To test whether the number of adverse events was different in the two groups, a two-sample Poisson test was used.

## 5. Conclusions

Successful ADSC-related improvement of wound healing correlated with the molecular profile, shedding light on the principles of cell-based therapy for DFU.

The encouraging clinical outcomes warrant further investigation of ADSC-based DFU treatment.

The accessibility of the material of treated wound sites at successive observation timepoints, as done here, presents a unique opportunity for further research on controlling the healing process and understanding the delivered cell-related pathway.

Given the state of the art so far, where the mode of action of cells applied for therapy has been elucidated mainly by in vitro studies and animal experiments, such a possibility might be a fundamental step toward realizing cell-based therapies for humans.

## Figures and Tables

**Figure 1 ijms-24-01472-f001:**
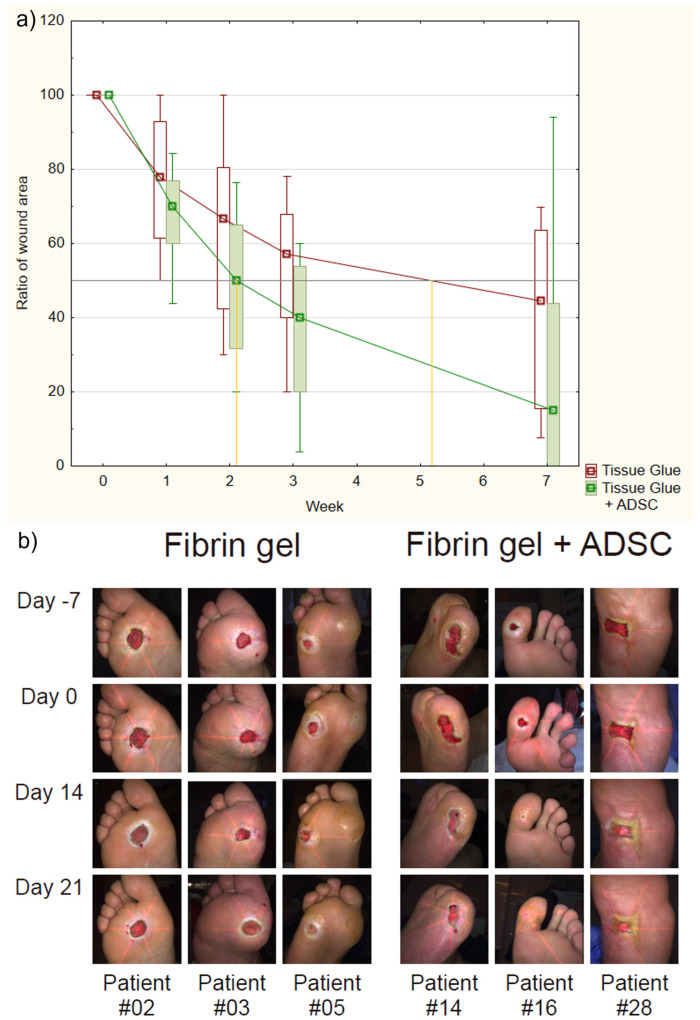
Comparison of fibrin gel group and ADSC group patients’ course of treatment: (**a**) Time dependence of the relative size of the wound in both groups. Relative wound size is defined as the ratio of the wound size at a given week to the size at week zero. At week zero, patients received either tissue glue or tissue glue along with ADSC; (**b**) Photographs of wound areas in the studied groups illustrating the course of DFU treatment in the study and the difference between patients from the fibrin gel group and the ADSC group. Photographs were taken in weekly intervals starting from the screening visit (Day-7), treatment application (Day 0), and two and three weeks into the treatment (Day 14 and Day 21, respectively).

**Figure 2 ijms-24-01472-f002:**
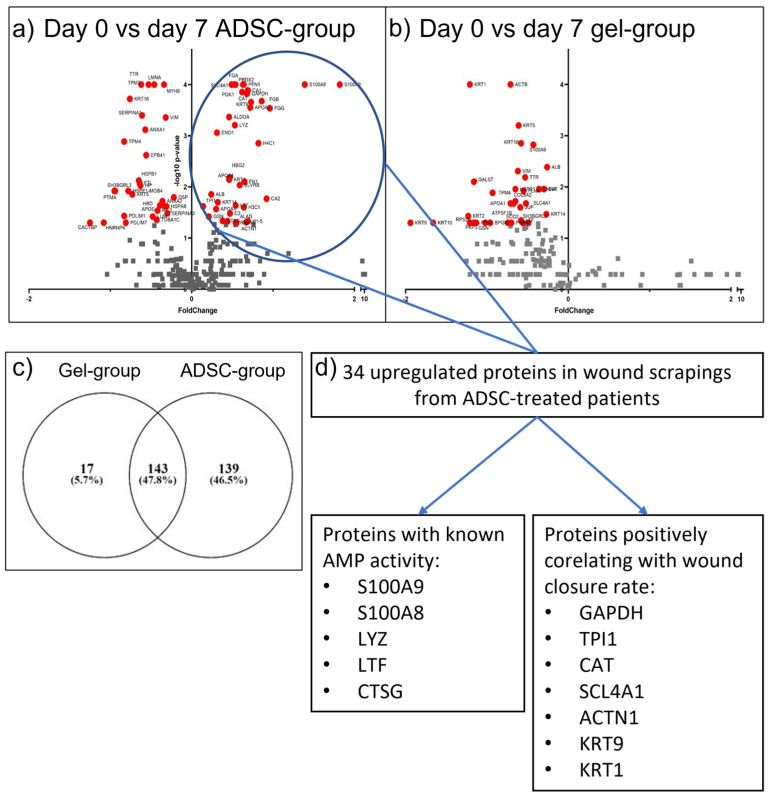
Results of wound scrapings proteomic analysis—Volcano plot of the identified protein intensities: (**a**) ADSC group day 0 vs. day 7; (**b**) Fibrin gel group day 0 vs. day 7; (**c**) Venn diagram comparing identified proteins in the ADSC group and fibrin gel group in both day 0 and day 7; (**d**) Position at volcano-plot and highlighted antimicrobial proteins and proteins correlating positively with wound healing.

**Figure 3 ijms-24-01472-f003:**
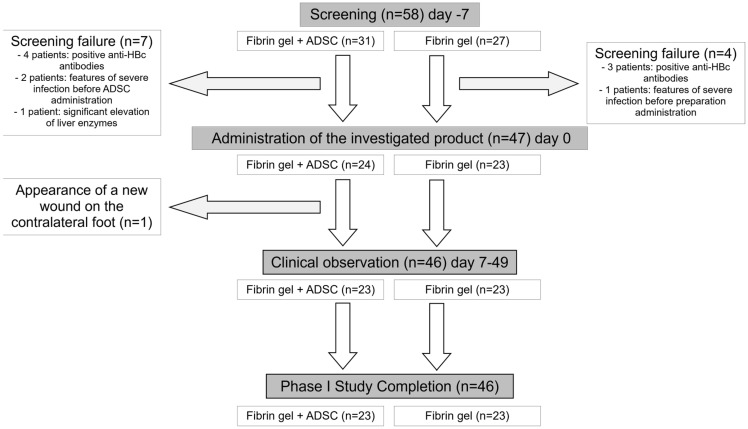
A consort diagram illustrating the flow of patients included in the study. Gray boxes represent the main stages of the study, with the total number of patients at the respective study point; vertical arrows illustrate the flow of patients between them. White frames below show the number of individuals in the respective treatment arms (“Fibrin gel + ADSC” or “Fibrin gel” alone), at each study stage. The drop-out cases are shown in boxes lateral to the horizontal arrows.

**Table 1 ijms-24-01472-t001:** Group characteristics.

		ADSC (N = 23)	Fibrin Gel (N = 23)	Total (N = 46)	*p* Value
**Gender (N, %)**	M	20 (87%)	17 (74%)	37 (80%)	0.459
	F	3 (13%)	6 (26%)	9 (20%)	
**Age (years)**	Mean ± SD	56.7 (11.1)	61.7 (7.5)	59.2 (9.7)	0.130
	Median	56.0	63.0	61.0	
	Min, Max	42.0, 75.0	38.0, 74.0	38.0, 75.0	
**Body weight (kg)**	Mean ± SD	97.0 (20.4)	103.6 (18.2)	100.3 (19.4)	0.395
	Median	98.0	105.0	101.0	
	Min, Max	49.0, 130.0	68.0, 140.0	49.0, 140.0	
**Height (cm)**	Mean ± SD	179.7 (9.2)	177.1 (7.6)	178.4 (8.5)	0.499
	Median	178.0	179.0	178.0	
	Min, Max	160.0, 200.0	160.0, 188.0	160.0, 200.0	
**Body mass index (kg/m^2^)**	Mean ± SD	30.0 (5.9)	33.0 (5.6)	31.5 (5.9)	0.068
	Median	30.8	33.0	31.6	
	Min, Max	19.1, 39.8	21.5, 49.6	19.1, 49.6	
**Wound size in week 0 (cm^2^)**	Mean ± SD	2.72 (2.85)	2.68 (1.58)	2.70 (2.28)	0.420
	Median	1.9	2.20	1.95	
	Min, Max	1.0, 14.0	1.00, 5.6	1.00, 14.00	
**Ulcer location (N, %)**	Left limb	11 (48%)	11 (48%)	22 (48%)	1.000
	Right limb	12 (52%)	12 (52%)	24 (52%)	
	Forefoot	18 (78%)	16 (70%)	34 (74%)	0.099
	Hindfoot	0 (0%)	4 (17%)	4 (9%)	
	Midfoot	5 (22%)	3 (13%)	8 (17%)	
	Plantar	17 (74%)	20 (87%)	37 (80%)	0.486
	Dorsal	3 (13%)	2 (9%)	5 (11%)	
	Both	3 (13%)	1 (4%)	4 (9%)	
**Duration of diabetic foot ulcers (months)**	Mean ± SD	19.5 (31.1)	30.2 (35.9)	24.9 (33.6)	0.155
	Median	9.0	14.0	12.0	
	Min, Max	1.0, 144.0	1.0, 156.0	1.0, 156.0	
**Duration of diabetes mellitus (years)**	Mean ± SD	16.1 (8.6)	19.7 (10.7)	17.9 (9.8)	0.424
	Median	16.0	18.0	16.0	
	Min, Max	2.0, 39.0	3.0, 44.0	2.0, 44.0	
**DM type (n, %)**	Type 1	4 (17%)	5 (22%)	9 (20%)	0.361
	Type 2	15 (65%)	17 (74%)	32 (70%)	
	Other	4 (17%)	1 (4%)	5 (11%)	
**Neuropathy (N, %)**	Yes	23 (100%)	23 (100%)	46 (100%)	---
	No	0 (0%)	0 (0%)	0 (0%)	
**Retinopathy (N, %)**	Yes	14 (61%)	13 (57%)	27 (59%)	0.811
	No	9 (39%)	10 (43%)	19 (41%)	
**Coronary artery disease (N, %)**	Yes	3 (13%)	6 (26%)	9 (20%)	0.459
	No	20 (87%)	17 (74%)	37 (80%)	
**Hypertension (N, %)**	Yes	16 (70%)	22 (96%)	38 (83%)	0.130
	No	7 (30%)	1 (4%)	8 (17%)	
**Smoking history (N, %)**	Yes	15 (65%)	14 (61%)	29 (63%)	0.811
	No	8 (35%)	9 (39%)	17 (37%)	
**ABI**	Mean ± SD	1.05 (0.14)	1.11 (0.18)	1.08 (0.16)	0.130
	Median	1.00	1.08	1.00	
	Min, Max	0.83, 1.40	0.70, 1.45	0.70, 1.45	
**tPO2**	Mean ± SD	54.6 (19.3)	52.3 (17.9)	53.4 (18.4)	0.695
	Median	53.0	50.0	50.5	
	Min, Max	25.0, 98.0	25.0, 87.0	25.0, 98.0	
**Glycated hemoglobin** **A1c at screening visit** **(%)**	Mean ± SD	7.25 (1.31)	7.57 (1.35)	7.41 (1.33)	0.536
	Median	7.40	7.30	7.40	
	Min, Max	5.00, 10.70	5.40, 10.80	5.00, 10.80	
**Creatinine level (mg/dL)**	Mean ± SD	1.11 (0.28)	1.17 (0.38)	1.14 (0.33)	0.828
	Median	1.11	1.05	1.07	
	Min, Max	0.64, 1.60	0.74, 1.87	0.64, 1.87	
**GFR (mL/min/1.73 m^2^)**	Mean ± SD	74.8 (21.2)	71.3 (24.4)	73.0 (22.7)	0.811
	Median	78.0	74.0	76.0	
	Min, Max	45.0, 113.0	27.0, 105.0	27.0, 113.0	

Data for patients’ clinical parameters mean (SD), median, minimum and maximum. In the last column are the results of the Wilcoxon test (for quantitative variables) and Fisher’s exact test (for categorical variables) comparing the ADSC and the fibrin gel groups. None of the differences fulfilled the *p* < 0.05 cutoff point. DM—diabetes mellitus; ABI—ankle brachial index; GFR—glomerular filtration rate; tPO2—transcutaneous oxygen tension.

## Data Availability

All data generated or analyzed during this study are included in the published article and its online Appendix A.

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
