# Peer review of "Allogenic Adipose-Derived Stem Cells in Diabetic Foot Ulcer Treatment: Clinical Effectiveness, Safety, Survival in the Wound Site, and Proteomic Impact"

_ijms, 2023, doi:10.3390/ijms24021472_

Round 1

Reviewer 1 Report

Dear authors.

The paper presented for review is very interesting and it addresses a very important problem of wound healing in type II diabetes. The problem it addresses concerns a study on the use of advanced therapy medicinal products. The results obtained are very interesting and are much better than those of the already registered product Alofisel, whose efficacy is much lower. Of course, this product has a different purpose but is based on the same type of cells.

1 Introduction. The statement concerning the "extraction of cells from umbilical cord blood" should be made more specific, as the use of this source in therapy is often the subject of fraud. It should also be mentioned that, despite numerous studies, there is only one adipose tissue stem cell preparation registered in the EU.

2. Information regarding the preparation itself should be detailed - a description of what the cells were suspended in - the presence of bovine serum, in which medium, method of isolation. The description in the publication does not allow an assessment of what exactly was used in the treatment. The fibrin gel also needs to be described - composition, ready-made, or prepared in an ATMP factory (how).

3. How many stem cells were in the preparation, where they sorted, and what were the evaluation parameters used E.g. CD44 only, or others. 

Only the total number of all cells is given in the text. 

4. Why cells from only one donor were used in the study.

5. Photographic documentation from other patients should be completed in the supplement.

Author Response

Dear Sir, Madame,

Thank you very much for your opinion on our manuscript.

Please, find  the detailed response to your comments in the attached file.

Reviewer 2 Report

The study evaluated the clinical effectiveness and safety of allogenic adipose-derived stem cells (ADSCs) in diabetic foot ulcer treatment. The article is important since allogenic but not allogenic ADSCs were used to treat foot ulcer in DM patients. Promising findings were shown. However, there are some concerns shall be verified. Other issues that need to be clarified/addressed are outlined below.

Abstract,

Page 1, line30-34,

“The clinical effects were assessed at 4 time points.”. The precise “4” time points shall be provided (w 1, 2, 3, and 7). Since the author reported that “The healing time was significantly shorter with greater reduction in wound size at subsequent visits in patients who received ADSCs. Complete healing was achieved in 7 patients treated with ADSCs vs. 1 treated without ADSCs.”. There is no info regarding the time points. Especially, there was no significant difference in the reduction in the size of the wound in the first- and second-weeks post-OP. Therefore, the time points of subsequent visits and complete healing shall be provided.

Page 1, line 33-34,

“7 of which positively correlated with the healing rate.” The authors shall list these critical proteins in the abstract. Or it is difficult to know the “molecular background” of the role of allogenic ADSCs in DM wound healing.

As I know, the abstract should be a total of about 200 words maximum to IJMS. However, the authors shall still try to provide more detail info in the abstract.

Keywords,

Page 1, line 38,

What is ATMP in the keywords? “ADSC” just represented the abbreviation “adipose-derived stem cells”, which is not necessary. Otherwise, “allogenic transplantation” is more important to this article and shall be included in the keywords.

Introduction

Page 2, line 49-51,

“Adipose-derived stem cells (ADSCs) are one of the most common materials used for MSC therapy, as they are abundant, easy to obtain and have high imunomodulatory potency [4, 9-12].”.

For ref 11, tissue regeneration capacity of EVs isolated ADSCs was discussed, which may not be suitable to be cited here.

Page 2, line 54-56,

Since this study is inspired by ref 14 “Allogeneic ASC sheet (ALLO-ASC-Sheet; Anterogen, Seoul, South Korea) is a 5 × 5 cm hydrogel sheet containing allogeneic ASCs.”. Results of ref 14 shall be summarized in detail. In addition, in ref 14, there was no detail info for the material of their “hydrogel sheet”. If the authors can provide/clarify more detail info to this issue, it may benefit the comparison of these 2 studies.

Results

Page 2, line 88,

“Evaluation of pain sensation did not differ significantly between the groups (Table S1).” Which reveals the ADSCs with/out fibrin gel may not improve/influence nerve regeneration and could be discussed.

In addition, the time points in Table S1 is week -7, 0, 1, 2, 3, and 4. I assumed week -7 shall be day -7 or week -1. For the 4 assessed time points, it shall be week 1, 2, 3, and 7. However, week 1, 2, 3, 4 were shown in Table S1. Which one is correct?

Page 4, section 2.2,

The time points V0 and V1 were referred to d-7 and d0 or d0 and d7? It is recommended to use the same terms such as d0, d7 but not V0, V1 (also Q1, Q2 were used in the Table S4). Or, the V1, V2 shall be defined in detail (such as the table captions in Table S5).

Page 4, line 116-117,

“Of all recipient wound scraping samples from the ADSC group, donor DNA was detected in 8 and 9 samples using STR and ADS, respectively.”

What are 8 and 9 samples? No. 8 and 9 patients? However, the following sentence described that “Donor DNA was mainly (7/21) found at the V1 time point”, which made reviewer/reader confusing.

Page 4, line 119-121,

“Donor DNA was mainly (7/21) found at the V1 time point, but single samples were positive at time points V2 and V3.” What were “single” samples meaning?

In addition, in line 121, “In the two scraping samples where donor DNA was detected at V2 and V3…”. The positive sample at V3 (1ABC01027) is negative at V2 (Table S5), which shall be highlighted and explained in the main contexts.

Furthermore, for 1ABC01019 at V2 and 1ABC01027 at V3, the amount of donor DNA was higher than V1, why?

Again, V1 represents d7? Since only single sample was positive at V2 and V3 (also in different subject), the long-term effects of ADSCs on DM foot ulcer shall be discussed.

Discussion

Page 7, line 171-172,

“Both methods showed that the DNA traces of donor cells were found even 21 days after administration of the medicinal product.”

However, only one sample was positive at V3, which shall be emphasized (1ABC01027). Also the line 175-179.

Page 7, line 179-180,

“Although we found ADSC donor genetic traces in only approximately 30% of patients at different time points”

It shall be emphasized the time point is V1 here.

Page 7, line 183-184,

“it is possible that ADSCs accumulated in healthy tissue that had already undergone epithelialization.”

The statement is an assumption. There is no solid evidence to support this statement.

Page 7, line 185-186,

“We identified 34 proteins that were highly overrepresented or identified exclusively in samples from day 7 after ADSC application compared to day 0 (before application).”

However, V0 and V1 were used in section 2.2. The authors shall use the same terms for time points.

Page 7, line 194-195,

“These proteins are involved in macrophage polarization from the M1 to the M2 phenotype and in the inflammation-to-proliferation phase shift in wound healing”

However, for ref 21, the CAT protein is related to ROS elimination/inflammatory stage arresting but not M1/M2 macrophage polarization.

Materials and methods

Page 8, section 4.2. ADSC collection and preparation,

There is no characterization for the ADSCs. The surface markers as well as differentiation potentials of hADSCs shall be characterized.

Page 9, section 4.3. Clinical procedures

The time points of V1, V0 etc shall be clarified. It is still recommended to use the same terms for time points (line 290-291).

Page 9, line 284,

How to prepare the fibrin gel shall be explained.

Page 9, line 285,

What is the “special applicator”? How to deliver the ADSCs/fibrin gel shall be described in detail.

Conclusions

Page 11, line 372-373,

“The encouraging clinical outcomes of both ours and Moon’s studies warrant further investigation of ADSC-based DFU treatment.”

The outcome of Moon’s study is not the finding of this study and shall be excluded from the conclusion.

References

For ref 2, it is a new paper without page number (online first). It is recommended to add the doi for this reference.

The page number for ref 8, 21, 25, 46, 51, and 54 is missing.

Author Response

Dear Sir, Madame,

Thank you very much for your opinion on our manuscript.

Please, find  below the detailed response (bolded) to your comments:

Comments and Suggestions for Authors

The study evaluated the clinical effectiveness and safety of allogenic adipose-derived stem cells (ADSCs) in diabetic foot ulcer treatment. The article is important since allogenic but not allogenic ADSCs were used to treat foot ulcer in DM patients. Promising findings were shown. However, there are some concerns shall be verified. Other issues that need to be clarified/addressed are outlined below.

Abstract,

Page 1, line30-34,

“The clinical effects were assessed at 4 time points.”. The precise “4” time points shall be provided (w 1, 2, 3, and 7). Since the author reported that “The healing time was significantly shorter with greater reduction in wound size at subsequent visits in patients who received ADSCs. Complete healing was achieved in 7 patients treated with ADSCs vs. 1 treated without ADSCs.”. There is no info regarding the time points. Especially, there was no significant difference in the reduction in the size of the wound in the first- and second-weeks post-OP. Therefore, the time points of subsequent visits and complete healing shall be provided.

Following the Reviewer’s suggestions we used uniform references to time points, i.e. day ‑7 (screening), day 0 (fibrin gel with ADSCs or fibrin gel alone), day 7, 14, 21 and 49 (control visits) in the whole text. The information about the significance of the changes in relation to the time points has been included in the abstract – lines: 28-29. Thanks to the unification of the description of time points it is also more clear in the Results – lines: 90-92.

The complete healing was achieved at the end of the study – included in the abstract – line 33 (also, line 93 in Results)

Page 1, line 33-34,

“7 of which positively correlated with the healing rate.” The authors shall list these critical proteins in the abstract. Or it is difficult to know the “molecular background” of the role of allogenic ADSCs in DM wound healing.

As I know, the abstract should be a total of about 200 words maximum to IJMS. However, the authors shall still try to provide more detail info in the abstract.

Done – lines: 35-36

Keywords,

Page 1, line 38,

What is ATMP in the keywords? “ADSC” just represented the abbreviation “adipose-derived stem cells”, which is not necessary. Otherwise, “allogenic transplantation” is more important to this article and shall be included in the keywords.

ATMP is an abbreviation for: Advanced Therapy Medicinal Products – line 317 in the text.

Working on the development of the new cell-based therapies in our daily practice, we often use this abbreviation for the literature search. Therefore,  we would prefer to leave it among the keywords. For the same reasons, we would prefer to leave the "ADSC" as a keyword, even if it is a repetition of a full term already used.

Especially, that there is still a space to add another keyword, which we have done following the Reviewer’s suggestion, with minor modification:

Instead of “allogenic transplantation” we have used “allogenic cell therapy”. This is because, according to the rules and regulations relating to cell therapies, such use of cells as in our case is not a form of transplantation, but it is the use of a cell-based medicinal product. The allogenic context is very important - we are very grateful for this suggestion.

Introduction

Page 2, line 49-51,

“Adipose-derived stem cells (ADSCs) are one of the most common materials used for MSC therapy, as they are abundant, easy to obtain and have high imunomodulatory potency [4, 9-12].”.

For ref 11, tissue regeneration capacity of EVs isolated ADSCs was discussed, which may not be suitable to be cited here.

Even though in ref. 11 only EVs capacity for tissue regeneration was discussed, it is still ADSCs that were the starting material for therapy. We would prefer to leave this citation also due to the fact that the use of ADSCs as a starting material to obtain a valuable derivatives is currently a very hot topic and we would not like to omit it completely, although a deeper discussion is beyond the scope of this publication. 

Page 2, line 54-56,

Since this study is inspired by ref 14 “Allogeneic ASC sheet (ALLO-ASC-Sheet; Anterogen, Seoul, South Korea) is a 5 × 5 cm hydrogel sheet containing allogeneic ASCs.”. Results of ref 14 shall be summarized in detail. In addition, in ref 14, there was no detail info for the material of their “hydrogel sheet”. If the authors can provide/clarify more detail info to this issue, it may benefit the comparison of these 2 studies.

Unfortunately, we cannot deliver more information about the hydrogel used by Moon et al. We could only speculate on that basing on several Korean patents. However the patent descriptions are very wide and the particular content of the hydrogel used in the study cannot be specified. Nevertheless, following the Reviewer’s suggestion, we expanded the discussion with a more direct comparison to the Moon's study – lines: 184-192 and 200-202. Thanks to this, the relevance of our results is more clearly indicated – therefore we are very grateful for this suggestion.

Results

Page 2, line 88,

“Evaluation of pain sensation did not differ significantly between the groups (Table S1).” Which reveals the ADSCs with/out fibrin gel may not improve/influence nerve regeneration and could be discussed.

In addition, the time points in Table S1 is week -7, 0, 1, 2, 3, and 4. I assumed week -7 shall be day -7 or week -1. For the 4 assessed time points, it shall be week 1, 2, 3, and 7. However, week 1, 2, 3, 4 were shown in Table S1. Which one is correct?

As indicated in the response to the first remark of the Reviewer, all the issues connected with the time points has been put into order. All changes are marked

 Page 4, section 2.2,

The time points V0 and V1 were referred to d-7 and d0 or d0 and d7? It is recommended to use the same terms such as d0, d7 but not V0, V1 (also Q1, Q2 were used in the Table S4). Or, the V1, V2 shall be defined in detail (such as the table captions in Table S5).

As indicated in the response to the first remark of the Reviewer, all the issues connected with the time points has been put into order. All numerous consistent changes are marked throughout the text.

Page 4, line 116-117,

“Of all recipient wound scraping samples from the ADSC group, donor DNA was detected in 8 and 9 samples using STR and ADS, respectively.”

What are 8 and 9 samples? No. 8 and 9 patients? However, the following sentence described that “Donor DNA was mainly (7/21) found at the V1 time point”, which made reviewer/reader confusing.

For response, see below

Page 4, line 119-121,

“Donor DNA was mainly (7/21) found at the V1 time point, but single samples were positive at time points V2 and V3.” What were “single” samples meaning?

In addition, in line 121, “In the two scraping samples where donor DNA was detected at V2 and V3…”. The positive sample at V3 (1ABC01027) is negative at V2 (Table S5), which shall be highlighted and explained in the main contexts.

Furthermore, for 1ABC01019 at V2 and 1ABC01027 at V3, the amount of donor DNA was higher than V1, why?

Again, V1 represents d7? Since only single sample was positive at V2 and V3 (also in different subject), the long-term effects of ADSCs on DM foot ulcer shall be discussed.

Regarding the two points above – “2.3. Donor DNA in wound samples”

This part of the text has been modified in order to clarify all the issues – both connected with the time points description and DNA sampling – changes marked in lines: 134-148.

Discussion

Page 7, line 171-172,

“Both methods showed that the DNA traces of donor cells were found even 21 days after administration of the medicinal product.”

However, only one sample was positive at V3, which shall be emphasized (1ABC01027). Also the line 175-179.

Done – line 206

 Page 7, line 179-180,

“Although we found ADSC donor genetic traces in only approximately 30% of patients at different time points”

It shall be emphasized the time point is V1 here.

Done – line 214

Page 7, line 183-184,

“it is possible that ADSCs accumulated in healthy tissue that had already undergone epithelialization.”

The statement is an assumption. There is no solid evidence to support this statement.

We have change the text following the Reviewer’s suggestion – lines: 216-230

Page 7, line 185-186,

“We identified 34 proteins that were highly overrepresented or identified exclusively in samples from day 7 after ADSC application compared to day 0 (before application).”

However, V0 and V1 were used in section 2.2. The authors shall use the same terms for time points.

As indicated in the response to the first remark of the Reviewer, all the issues connected with the time points has been put into order. All numerous consistent changes are marked throughout the text.

Page 7, line 194-195,

“These proteins are involved in macrophage polarization from the M1 to the M2 phenotype and in the inflammation-to-proliferation phase shift in wound healing”

However, for ref 21, the CAT protein is related to ROS elimination/inflammatory stage arresting but not M1/M2 macrophage polarization.

The more appropriate reference has been provided:

  1. Park, Ye Seul, Md Jamal Uddin, Lingjuan Piao, Inah Hwang, Jung Hwa Lee, and Hunjoo Ha. "Novel Role of Endogenous Catalase in Macrophage Polarization in Adipose Tissue." Mediators of inflammation 2016 (2016)

Materials and methods

Page 8, section 4.2. ADSC collection and preparation,

There is no characterization for the ADSCs. The surface markers as well as differentiation potentials of hADSCs shall be characterized.

Surface markers characteristics performed by FACS are described in the revised manuscript – lines: 339-346

Regarding the differentiation markers, in our cell bank we do not verify it for every single donor. We have a collection of ADSCs from over 100 donors – all isolated in the same manner as in this study, and we performed the differentiation tests toward osteogenic, chondrogenic and adipogenic immunophenotypes for a vast majority of them. Their differentiation potential is different in terms of the time in culture needed to obtain the terminally differentiated cell population. Nevertheless, they all respond to the differentiating factors in culture, as required by the minimal MSCs criteria.

 Page 9, section 4.3. Clinical procedures

The time points of V1, V0 etc shall be clarified. It is still recommended to use the same terms for time points (line 290-291).

As indicated in the response to the first remark of the Reviewer, all the issues connected with the time points has been put into order. For this particular issue – see lines: 363, and 368, and also other highlighted changes in this section.

Page 9, line 284,

How to prepare the fibrin gel shall be explained.

Done – lines: 371-379

 Page 9, line 285,

What is the “special applicator”? How to deliver the ADSCs/fibrin gel shall be described in detail.

Done – lines: 371-379

Conclusions

Page 11, line 372-373,

“The encouraging clinical outcomes of both ours and Moon’s studies warrant further investigation of ADSC-based DFU treatment.”

The outcome of Moon’s study is not the finding of this study and shall be excluded from the conclusion.

Done – line 468

References

For ref 2, it is a new paper without page number (online first). It is recommended to add the doi for this reference.

 Done

The page number for ref 8, 21, 25, 46, 51, and 54 is missing.

Ref. 21 has been changed (following the Reviewer’s suggestion – described above), the current cited publication has been published online-only. Thus we provided doi instead of page numbers. Ref.2, 51 and 54 have been supplemented with doi numbers, as for references 8, 25 and 46 the appropriate page numbers were provided.

Reviewer 3 Report

The authors of this work present a significant study regarding the local treatment of human diabetic foot ulcers with a cell product containing adipose tissue-derived allogeneic cells. They showed an improved healing rate in the study group compared to the control group  - fibrin gel administration only. Moreover, proteomic analysis was carried out from the wound scraping obtained from debridement procedures. This analysis revealed the over-presentation of several proteins, some of them being linked to wound healing. I can see the following strengths of the paper: up-to-date topic, novelty, clinical study, trial registration, very good presentation and level of English, use of donor allogeneic cells that spare the patient from a harvesting procedure, and insight into the mechanism of action of the cell therapy product. Limitations are recognized, and they include a nonrandomized study. I recommend publishing the main manuscript in its current form. The supplement material, however, would benefit from formal changes, such as uniform script type and size, including data in the Tables and line spacing of the text. These minor issues will probably be addressed by a journal text editor anyway. 

Author Response

Dear Sir/Madame,

We are very grateful for your very positive opinion.

Regarding the formal changes in the supplementary material, we will be happy to make all the required changes in a colaboration with a journal text editor, especially because of the diversity of the type of the materials included.

Very many thanks, once again,

Sincerely, on behalf of the authors,

Malgorzata Lewandowska-Szumiel

Round 2

Reviewer 1 Report

Dear authors,

thank you for your comments and corrections. I wish you continued success in your stem cell research. 

Author Response

Very many thanks!

Sincerely,

Malgorzata Lewandowska-Szumiel

Reviewer 2 Report

This draft of manuscript is well revised. Some minor questions could be further improved.

Line 102-104,

“analysis of the SDF-1/CXCR4 pathway”

A ref is required here.

For ref 21, the page number (article number) can be found.

Mediators Inflamm. 2016;2016:8675905.

Ref 52, Biomed Res Int. 2019; 2019:2626374.

Ref 55, Clin Dev Immunol. 2012;2012:534291.

Author Response

Dear Sir/Madame,

Thank you for your comments.

Please, find the response in the enclosed file.

Sincerely Yours,

Malgorzata Lewandowska-Szumiel
